# Differences in Troponin I and Troponin T Release in High-Performance Athletes Outside of Competition

**DOI:** 10.3390/ijms25021062

**Published:** 2024-01-15

**Authors:** Jan C. Wuestenfeld, Tom Kastner, Judith Hesse, Leon Fesseler, Florian Frohberg, Cornelius Rossbach, Bernd Wolfarth

**Affiliations:** 1Institute for Applied Training Science, Marschnerstrasse 29, 04109 Leipzig, Germany; 2Charité—Universitätsmedizin Berlin, Corporate Member of Freie Universität Berlin and Humboldt Universität zu Berlin, Department of Sports Medicine, Charitéplatz 1, 10117 Berlin, Germany

**Keywords:** troponin I, troponin T, sport, elite athletes, exercise, cardiac health

## Abstract

Troponin I and troponin T are critical biomarkers for myocardial infarction and damage and are pivotal in cardiological and laboratory diagnostics, including emergency settings. Rapid testing protocols have been developed for urgent care, particularly in emergency outpatient clinics. Studies indicate that strenuous physical activity can cause transient increases in these troponin levels, which are typically considered benign. This research focused on 219 elite athletes from national teams, evaluating their troponin I and T levels as part of routine sports medical exams, independent of competition-related physical stress. The results showed that 9.2% (18 athletes) had elevated troponin I levels above the reporting threshold, while their troponin T levels remained within the normal range. Conversely, only 0.9% (two athletes) had normal troponin I but raised troponin T levels, and 2.3% (five athletes) exhibited increases in both markers. No significant cardiovascular differences were noted between those with elevated troponin levels and those without. This study concludes that elevated troponin I is a common response to the intense physical training endured by high-performance endurance athletes, whereas troponin T elevation does not seem to be directly linked to physical exertion in this group. For cardiac assessments, particularly when ruling out cardiac damage in these athletes, troponin T might be a more reliable indicator than troponin I.

## 1. Introduction

Moderate physical activity is generally considered to be healthy and is associated with a number of benefits [1,2]. It is well established that exercise can elevate cardiac troponin (cTn) levels [3,4,5,6]. Studies on marathon runners, ergometer tests, and ultramarathon runners have repeatedly shown that exercise results in elevations in blood cTn [3,6,7]. Although the pathogenesis of exercise-induced cTn release is not fully understood, its pathological relevance is generally considered low or irrelevant, a view not universally accepted [6]. The degree of troponin elevation post-exercise varies significantly among individuals, influenced by factors like training status, age, exercise duration, and sex [4]. However, no data have been published on the differences between different causes of troponin elevation in response to exercise. Furthermore, whether there is a difference in the exercise-induced increase in troponin I (cTnI) and troponin T (cTnT) is not yet known.

The literature suggests that exercise-induced troponin elevation is more prevalent in men than women [8] and occurs more frequently in young and middle-aged athletes [7,9,10]. Prior intense exercise has been shown to significantly affect serum cTn levels, leading to increased detection of both cTnT and cTnI. However, most studies have not differentiated between the two troponins, often focusing on just one isoform [4,9,11,12,13]. This paper, therefore, aims to show the extent to which physical training activity in high-performance athletes has an impact on the detectability of both cTnI and cTnT.

The reasons behind the detectability of troponins in the blood post-exercise are not yet fully understood. Katus et al. found no explanation for elevated resting cTnT values following 12 weeks of HIIT and CAT training [14], but they were reproduced by Legaz-Arrese et al. [9]. In addition, higher post-exercise cTnT levels were found to be associated with increasing exercise capacity without a strong association between the two parameters. One theory for these data is that adaptive processes in the heart as a result of training explain the higher cTn values. Contrary to this, Nie et al. showed in two papers that 12 weeks of training led to lower post-exercise cTnT values [13]. The author concluded that this could be due to reduced mechanical load on the heart as it adapts to the training stimulus. This conclusion can be drawn from the lower heart rate during exercise and the shorter exercise duration. The training probably did not lead to the lower post-exercise cTn values but rather shifted the absolute exercise threshold for troponin elevations due to the improved fitness of the subjects [13]. No correlation could be found between previous exercise experience and post-exercise troponin levels. Eijsvogels et al. [7] recognized in marathon runners that older age correlates with training experience and exercise intensity, but their statistical model isolated training experience and showed that this was not a relevant factor for the level of post-exercise cTn values.

The differences in the exercise-induced increases in cTnI and cTnT have not been fully elucidated so far. If apparent differences between stress-induced increases in cTnI and cTnT are shown, this must be considered when diagnosing ACS, myocardial infarction, or myocarditis. To date, no distinction has been made between cTnI and cTnT [15]. This study was conducted to enhance our understanding of troponin increases due to physical activity, particularly to highlight any differences in the rise of various troponins among high-performance athletes.

## 2. Results

In this study, 219 high-performance athletes, averaging 23.7 years old (female: 23.8 years, male: 23.6 years) from 36 different sports (see Appendix A), were analyzed. The majority, particularly from endurance sports (total: 154, males: 85, females: 60), underwent resting ECGs that typically showed non-pathological results, with sport-specific changes in some cases. With a high proportion of top athletes from endurance disciplines, this corresponds to the expected results. The mean heart rate of 56.4 bpm across all athletes was slightly bradycardic. There was no significant PQ time prolongation over 200 ms or signs of atrioventricular shortened conduction. However, the QRS duration was slightly prolonged, averaging 106 ms across genders and notably longer in males (112.7 ms), suggesting physiological changes related to intensive training (see Table 1). Echocardiographic assessments revealed no pathological abnormalities, with many athletes showing expected sport-related cardiac adaptations, particularly noted in the relative heart size as per Dickhuth’s formula [16]. Male athletes generally had a slightly larger heart size relative to their body weight compared to females. Other parameters, including the diameter of the intraventricular septum, left ventricular internal diameter in diastole, and left atrial diameter in atrial diastole, were within the upper normal range or slightly increased, indicating typical athletic heart adaptations (see Table 2).

Inz the cTnT and cTnI analyses performed on the elite athletes, a total of 18 athletes (8.2%, 12 females, 6 males) were found to have elevated cTnI values and normal cTnT values. An increase in troponin levels was found exclusively in endurance sports (biathlon (eight), middle-distance running (two), cross-country skiing (seven), and triathlon (one)). With an average troponin increase of 92.6 ng/mL, this value was approximately double the upper standard limit, whereas the maximum cTnI increase of 399.5 ng/mL corresponds almost tenfold to the upper standard limit. In the differentiated analysis of the results of both sexes, no significant differences were found concerning the troponin mean values and maximum measured values. However, with 12 female athletes showing an elevated cTnI value, there were twice as many women in this group as men (6 athletes). While cross-country skiers with elevated cTnI values were found in both genders, this was only the case for female athletes in biathlons. In contrast, only two male athletes with elevated cTnI levels were found in middle-distance running (see Table 3).

In five athletes (2.3%, two females and three males), elevated values for cTnT and cTnI were found. The mean cTnI value of 528.2 was found to be increased by a factor of 11.7. Although this is almost ten times higher compared to the group of athletes with an exclusive cTnI increase, it is due to the maximum measured value of 967.8 ng cTnI of one athlete, which leads to this increase in the mean value. This maximum cTnI value of a female athlete (track and field athlete) corresponds to a 21.4-fold increase in the upper normal value. Similarly, a differentiated analysis of both sexes in the group of male athletes shows one athlete (a biathlete) with a 17.3-fold increase in the upper normal range for cTnI. While the other two male athletes showed comparatively low cTnI increases (61.2 ng/mL and 71.55 ng/mL, respectively), the cTnI increases of the female athletes, who also showed an increase in cTnT, were significantly higher (784.89 ng/mL and 967.8 ng/mL). The increase in cTnT found in this group of athletes was very moderate for all athletes. With a mean troponin value of 19 ng/mL, this value was only increased by a factor of 1.4 above the upper normal range. Furthermore, with a maximum cTnT increase of 21 ng/mL, this value was only 1.5 times higher than the upper normal range. The differentiated analysis of females and males showed no significant differences in the cTnT measured values with an overall small group size (both absolute group size and gender-related group size). In contrast to the group of athletes with only cTnI elevation, the group with cTnI and cTnT elevation included one athlete from a duel sport (wrestling) in addition to the athletes from endurance disciplines (see Table 4).

In two athletes (0.9%, one female, one male), elevated values for cTnT were found with normal values for cTnI. The female athlete with elevated cTnT and normal cTnI is a known carrier of Duchenne–Becker disease and was repeatedly found to have solitary elevated cTnT during the course of measurements. Since patients with Duchenne–Becker muscular dystrophy are known to have a permanent troponin elevation in the blood, according to the literature [17], it can be assumed that the elevation of cTnT in this female athlete is not due to competitive sports training but rather to an underlying genetic predisposition. With a value of 18 ng/mL (the corresponding cTnI value was 5.56 ng/mL), the male athlete with an exclusive cTnT increase shows a slight increase in cTnT of 1.3 times the upper normal value. However, it must also be pointed out that this is, again, an athlete from an endurance sport (cross-country skiing) (see Table 5).

No significant differences in anthropometric, echocardiographic, or ECG data were observed among the different groups categorized by their troponin levels (1. normal cTnI/cTnT, 2. elevated cTnI with normal cTnT, 3. normal cTnI with elevated cTnT, and 4. elevated cTnT and cTnI) (see Table 6).

Finally, it was statistically analyzed whether there is a correlation between cTn increase and physical findings such as heart rate, LV mass, etc. The Spearman correlation analysis between the values of troponin I and the other variables shows a slight to moderate negative correlation with troponin I, but none of the p-values are low enough to indicate a statistically significant correlation, except for IVSd. IVSd shows a moderate negative correlation (−0.22) with troponin I, and the p-value (0.0042) suggests a statistically significant correlation. The Spearman correlation analysis between troponin T and various variables predominantly showed non-significant correlations with slight to moderate negative or positive tendencies. Notably, body weight, LBM, and several other measurements indicated no correlation. A few variables, like body fat mass, exhibited a stronger positive correlation, yet the results were just above the threshold for statistical significance. Overall, no clear or statistically significant relationships were established, indicating a generally weak association between troponin T and the measured variables in this dataset. Due to the lack of statistically significant correlations, further elaboration on this matter in the paper is not pursued (compare Table 7).

## 3. Discussion

The phenomenon of increased troponin levels post-exercise is well documented in the literature [18], with athletes typically exhibiting mean elevations in the lower pathological range compared to individuals with cardiovascular diseases. Despite this knowledge, the precise mechanisms behind exercise-induced troponin release remain unclear. Various theories have been proposed, such as increased cardiomyocyte sarcolemmal permeability, enhanced turnover, and rates of apoptosis and necrosis [19]. However, due to assay specificity, the likelihood of cross-reactivity from muscle damage contributing to raised cTn concentrations is minimal [19,20].

While generally considered non-pathological, isolated instances suggest a potential link between exercise-induced troponin release and future cardiovascular events in older athletes [19]. Some researchers propose that athletes who experience troponin increases due to training stress might have a heightened long-term cardiovascular risk [21,22], though this remains speculative without longitudinal data. This study is distinctive in its focus on stress-induced troponin increases in elite high-performance athletes, providing valuable insights into this unique group. The troponin levels observed here are consistent with those reported in recreational and ambitious amateur athletes [20].

A review by Cirer-Sastre et al., among others, demonstrated significant post-exercise troponin increases. The upper reference limit was exceeded by 76% of participants for cTnT, 51% for cTnI, and 13% for NT-proBNP. Furthermore, the cutoff value for acute myocardial infarction was exceeded by 39% for cTnT and 11% for cTnI [23]. Corresponding values were also occasionally found in the data of the analyzed athletes. However, we were only able to demonstrate a correspondingly high increase in the measured values in the analyzed clientele for cTnI. When looking at the athletes in whom an elevated troponin value was detectable, it is noticeable in relation to the population of all study participants that almost exclusively endurance athletes had elevated troponin values. This fact has already been described by Tesema et al. in a study of amateur endurance athletes, in which a 12-week endurance training program was shown to be associated with cTnI and cardiovascular adaptation [24]. Similar results were described in a study of amateur triathletes observed over a more extended training period. While there was a correlation between the duration since the last training session and positive cTnI values, 20% of the blood samples examined from 15 athletes showed at least one elevated cTnI value. In three athletes, elevated cTnI was even detected in every blood sample [25]. However, almost all of the endurance athletes were in the first training period of the current training year when the blood sample was taken, as the annual basic sports medicine examinations typically take place during this phase. This training period is primarily characterized by a high proportion of low-intensity training with long training durations in all endurance sports. However, high-intensity training impulses are also occasionally used in this phase, so it is impossible to differentiate with certainty whether the proven troponin increases are caused by the low-intensity training with occasional very long training periods or rather the high-intensity interval training. In most published studies, however, either cTnI or cTnT have been examined. A direct comparison of the troponin elevations of cTnI and cTnT, especially in high-performance athletes, has not been performed so far. The data presented in this paper show a trend towards a more common exercise-induced increase in cTnI above the normal range in high-performance athletes during the training process than an increase in both troponins (I and T) or an increase in cTnT alone. Although the statistical power of the available data does not support this hypothesis, further studies should investigate whether there is a statistically significant increase in cTnI over cTnT in elite athletes, in contrast to many studies of exercise-induced troponin elevation in amateur athletes. In addition to the data presented on two athletes with an exclusive cTnT elevation, it must be mentioned that the female athlete is a known carrier of Duchenne–Becker’s disease. In these patients, troponin elevation due to the disease is very common, so the training load of the troponin elevation can probably be assumed to be non-causal. Nonetheless, the observed differences in troponin expression—only cTnI, only cTnT, or an increase in both troponins—remain unclear due to the underlying mechanism needing to be adequately elucidated. Although most authors do not consider exercise-induced troponin elevation to be pathological, it still cannot be ruled out that this abnormality could be a predictor of long-term cardiovascular disease, especially in younger athletes. For this reason, the athletes with abnormal troponin values were examined for differences in the cardiovascular examinations (resting ECG and echocardiography). These examinations revealed no evidence of an underlying disease. This confirms the statement by Jean-Gilles and Baggish that the significance of troponin determination in endurance athletes after physical exertion is very limited and, in particular, does not support triage decisions “at the finish line” [26]. Nevertheless, it is crucial to acknowledge that the daily training load for elite endurance athletes might induce troponin increases beyond clinical thresholds for myocardial ischemia. This presents a diagnostic challenge for physicians in distinguishing suspected myocardial damage in athletes outside of competition settings, as standard diagnostic algorithms lack specificity. Therefore, the findings should not preclude further diagnostics in athletes with clinical indications of myocardial damage.

## 4. Materials and Methods

### 4.1. Participants

This prospective study included elite high-performance athletes from an array of sports disciplines. Participants were enrolled between January 2021 and May 2022 and were part of their respective national selection squads or national teams. The athletes underwent their annual basic sports medical examination at our institute, primarily during their first preparation period of the training and competition year, which coincided with their regular training routines. Notably, no standardization based on prior training loads was performed during data collection. Therefore, the troponin data and cardiologic examination results reflect the athletes’ real-life training conditions.

In total, 219 athletes (124 male, 95 female) with an average age of 23.7 years (SD 4.7 years) participated in this study (refer to Table 8). Two athletes were excluded beforehand due to incomplete examination data. This study encompassed athletes from 36 different sports. Detailed information regarding the distribution of participants by sport and gender is available in the Appendix A Appendix A. All included athletes were healthy at the time of the examination and showed no signs of acute illness in the sports medical examination (medical history, clinical examination, laboratory diagnostics, resting ECG, echocardiography).

### 4.2. Troponin Tests

Venous blood samples were collected for serum analysis using 3.5 mL serum tubes (S-Monovette, 7.5 mL, Sarstedt, Nümbrecht, Germany). To quantitatively measure serum high-sensitivity cardiac troponin T (hs-cTnT), we utilized the cobas Elecsys Troponin T hs STAT assay from Roche Diagnostics GmbH, Mannheim, Germany, following the manufacturer’s instructions. The hs-cTnT assay has a limit of blank of 2.14 ng/L, a limit of detection of 5 ng/L, and a 99th percentile cutoff point for myocardial damage at 14 ng/L. The coefficient of variation (CV) for this assay is 7% at the cutoff level.

For the quantitative measurement of serum high-sensitivity troponin I (hs-cTnI), we employed the Atellica IM Analyzer (Siemens Healthineers AG, Erlangen, Germany) along with the High-Sensitivity Troponin I (TnIH) assay. This hs-cTnI assay features a limit of blank of 3 ng/L, a limit of detection of 5 ng/L, and a 99th percentile myocardial damage cutoff point of 45.43 ng/L. The coefficient of variation (CV) is ≤12% for samples with 9–20 pg/mL (ng/L) and ≤10% for samples >20 pg/mL (ng/L). The manufacturer’s upper standard range for this assay is 45 ng/mL.

### 4.3. Cardiac Screening

Each athlete in this study underwent a comprehensive internal medical history review and physical examination by an experienced sports physician. As part of their routine sports medical examination before testing, they also received a resting electrocardiogram (ECG), detailed anthropometric diagnostics, and a standardized echocardiographic examination. These procedures were performed per the recommendations provided by the German Society for Ultrasound in Medicine (DEGUM) and the German Society for Sports Medicine and Prevention (DGSP).

This study was carried out strictly following the ethical principles outlined in the Declaration of Helsinki. All participating athletes provided written informed consent for the data obtained from their sports medical examinations to be used anonymously for research purposes.

### 4.4. Statistical Analysis

An analysis of variance (ANOVA) was conducted to determine if there were significant differences in various parameters among the four distinct troponin groups. The groups were normal Trop I and T, elevated Trop I and norm. Trop T, norm. Trop I and elev. Trop T, and elevated Trop I and T. The analyzed parameters included age(years), body weight (kg), body height (cm), body fat mass (kg), body fat percentage (%), Body Mass Index (BMI), lean body mass (LBM) (kg), PQ time (ms), QRS time (ms), QTc time (ms), relative heart volume (HVrel) (HV/k), IVSd (cm), LA dimension (cm), LVIDd (cm), and resting heart rate (Rest HR).

## 5. Conclusions

In summary, this study underscores the complexity of interpreting cardiac troponin levels in high-performance athletes, particularly following endurance training. While both cTnI and cTnT are evidently influenced by physical exertion, the distinct nature of their responses and their interaction with various levels of endurance training remain less understood. These nuances highlight the need for further research to decipher the specific differences and underlying mechanisms. Such knowledge is essential for accurately interpreting troponin levels and making informed decisions regarding the cardiovascular health of athletes across a spectrum of endurance levels. As we continue to unravel these relationships, it will enhance our ability to effectively support athletes’ health and performance.

## Figures and Tables

**Table 1 ijms-25-01062-t001:** Baseline resting ECG data of all athletes.

ECG Duration (ms)	All	Female	Male
PQ Duration	146.20	144.72	147.33
SD	31.50	26.68	33.57
QRS Duration	106.3	97.9	112.7
SD	10.9	7.3	8.6
QTc time (Bazett)	402.08	419.67	406.26
SD	24.02	23.53	22.83
Mean heart rate	56.4	56.4	56.4
SD	10.5	11.7	9.6

**Table 2 ijms-25-01062-t002:** Baseline echocardiographic data of all athletes.

Value	All	Female	Male
Ejection fraction (%)	>65%	>65%	>65%
Heart volume/weight (mL/kg)	12.47	11.68	12.93
SD	2.03	1.72	2.12
Diameter interventr. Septum (iVSDd/cm)	1.01	0.94	1.07
SD	0.12	0.11	0.10
Left atrium (diameter cm)	3.67	3.43	3.85
SD	0.41	0.35	0.36
Left ventricle (diameter cm) LviDd	5.17	4.83	5.43
SD	0.47	0.32	0.40

**Table 3 ijms-25-01062-t003:** Athletes with elevated troponin I and normal troponin T values.

Value	All (18)	Female (12)	Male (6)
cTnI mean ng/mL	92.6	98.1	94.6
Min/Max ng/mL	46.2/399.8	46.2/399.8	53.2/321.7
Disciplines	Biathlon (8), middle-distance running (2), cross-country skiing (7), triathlon (1)	Biathlon (8), cross-country skiing (3), triathlon (1)	Middle-distance running (2), cross-country skiing (4)

**Table 4 ijms-25-01062-t004:** Athletes with elevated troponin I and elevated troponin T levels.

Value	All (5)	Female (2)	Male (3)
cTnI mean ng/mL	528.2	937.46	306.0
Min/Max ng/mL cTnI	61.2/967.8	907.12/967.8	61.2/784.9
cTnT mean ng/mL	19	18.3	20.5
Min/Max ng/mL cTnT	15/21	15/21	20/21
Disciplines	Biathlon (2), middle distance running (1), finswimming (1), wrestling (1)	Biathlon (1), middle distance running (1)	Biathlon (1), finswimming (1), wrestling (1)

**Table 5 ijms-25-01062-t005:** Athletes with normal troponin I and elevated troponin T levels.

Value	All (2)	Female (1)	Male (1)
cTnT mean ng/mL	20	22.0	18.0
Min/Max ng/mL cTnT	18.0/22.0	/	/
Disciplines	Cross-country skiing (2)	Cross-country skiing (1)	Cross-country skiing (1)

**Table 6 ijms-25-01062-t006:** Comparison of anthropometric, echocardiographic, and ECG values in athletes with different troponin levels.

	Normal cTnI and cTnT		Elev. cTnI, norm. cTnT		Norm. cTnI, elev. cTnT		Elevated cTnI and cTnT		
	Female	Male	Female	Male	Female	Male	Female	Male	*p*-Value
Age	24.33	23.81	21.42	21.83	21	21	21.0	20.3	n.s.
Height	169.90	185.19	170.62	183.58	166.4	185.2	165.0	186.87	n.s.
Weight	62.23	81.54	60.98	75.43	59.3	82.70	53.45	80.47	n.s.
BMI	21.86	23.66	20.94	22.35	21.4	24.1	19.70	23.00	n.s.
Fat %	14.48	11.24	14.89	10.0	14.8	9.7	12.0	10.5	n.s.
Fat mass	9.55	9.53	9.13	7.50	8.7	8.0	6.40	8.87	n.s.
LBM	53.69	72.01	51.86	67.93	50.60	74.70	47.05	72.07	n.s.
HV rel	11.77	12.90	12.34	13.86	11.60	14.84	12.82	11.85	n.s.
iVSd	0.93	1.07	0.95	1.03	0.80	1.07	0.98	1.03	n.s.
LViDd	4.83	5.42	4.91	5.44	4.89	5.78	4.32	5.45	n.s.
LA	3.42	3.87	3.53	3.62	3.83	4.0	3.40	3.57	n.s.
PQ time	144.77	146.87	158.0	145.17	119	174	155	157	n.s.
QRS time	98.0	112.3	97.7	119.8	101.0	107.0	94.0	115.3	n.s.
QTc time	420.38	406.07	412.75	400.33	430	421	428	420	n.s.
Rest HR	57.3	56.4	52.8	52.0	43	62	50	63.3	n.s.

**Table 7 ijms-25-01062-t007:** Statistical analysis.

Parameter	F-Value	*p*-Value
Age	2.59	0.054
Weight	1.69	0.170
Height	0.91	0.437
Fat Mass	0.45	0.715
Fat %	0.32	0.811
BMI	1.59	0.192
LBM	1.91	0.129
PQ Time	0.57	0.637
QRS Time	0.12	0.949
QTc Time	0.70	0.551
HV rel	0.34	0.797
IVSd	0.82	0.483
LA Dimension	0.99	0.398
LVIDd	0.53	0.665
Rest HR	0.94	0.423

**Table 8 ijms-25-01062-t008:** Baseline data of athletes.

	All	Female	Male
	*n* = 219	*n* = 95	*n* = 124
Age (years) SD	23.71	23.85	23.60
	4.66	4.50	4.79
Weight SD	73.28	62.70	81.23
	15.32	9.05	14.23
Height (cm)	9.93	169.85	185.15
SD	9.93	5.35	7.22
BMI	22.76	21.70	23.58
SD	3.03	2.64	3.06
Body fat percent (%) caliper	12.59	14.48	11.151
SD	4.49	11.15	3.20
Lean body mass (kg)	63.84	53.28	71.84
SD	12.58	5.36	10.39
Fat mass (kg)	9.41	9.42	9.40
SD	4.63	4.97	4.37

## Data Availability

All original data from this study are available to the authors in digital form and can be made available on request.

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
