# Peer review of "Differences in Troponin I and Troponin T Release in High-Performance Athletes Outside of Competition"

_ijms, 2024, doi:10.3390/ijms25021062_

Round 1
Reviewer 1 Report
Comments and Suggestions for Authors
Dear authors
I hope this letter finds you well. I would like to express my appreciation for your efforts in addressing a significant research question in your manuscript titled " Differences in Troponin I and Troponin T release in high performance athletes out of competition." I have carefully reviewed your work, and I commend your commitment to advancing knowledge in your field.
However, after thorough consideration, it is with regret that I must convey my concerns about the conceptual and design aspects of your manuscript. I believe there are significant errors that need to be addressed before the paper can be considered for publication. You can find further details at the bottom of this letter.
Thank you for your understanding and dedication.
Comments:
- The Introduction is excessively detailed for an original article. The focus of the study is missed and lacks a clear objective. Some references should be redistributed (in lines 51-52 the authors comment on variable sex and then again in line 61).
- “The screening phase was conducted according to the study protocol until a total of 20 athletes with elevated troponin I values above the upper reporting level (URL) and normal troponin T values could be identified. To do this, 219 athletes had to be included in the study”.
The authors should explain why they have selected this endpoint (20 athletes with cTnI > URL). Have you calculated the data size? Why not cTnT if in theory you did not know that that troponin was the one presenting elevation?
- If the design of the study consisted in including patients until you have 20 athletes with cTnI elevation, why in results you only have 18? Please comment on this.
- “While, in contrast to patients with cardiovascular diseases, the mean troponin elevations in athletes after exercise are comparatively low and in the lower pathological range, the underlying causes of troponin elevation after exercise are still not clear.”
Results on Table 6 cannot be considered as low. These findings have to be further discussed.
- “The data presented in this paper lead to the conclusion that an exercise-induced increase in troponin I above the normal range is much more common in high-performance athletes during the training process than an increase in both troponins (I and T) or an increase in troponin T alone.”
The study design does not allow this conclusion to be reached.
- A clear discussion of the authors’ results is missed.
- Line 58-59: Only Scherr et al [10] found higher exercise-induced cTnT values in the age group 29-51 years. Eijsvogels et al. suggested a relationship between …
The dot between years and Eijsvogels is missing.
- Line 62: difference for post exerciese cTnT,
Typo
- I suggest not to use the abbreviation “postex.”
- “Venous blood samples for plasma analysis were collected into 3,5ml Serum Plasma tubes,”
Please clarify if the tubes were serum or plasma. The manufacturer of the tube should be also declared.
- Table 2 could be presented as Supplemental Material.
- I recommend to use the abbreviations presented in line 50 along all the manuscript. In line 152 and Table 7, the authors use a new one (Trop T; Trop I)
Author Response
Dear reviewer,
Thank you very much for your detailed and extremely helpful comments on our manuscript, which we greatly appreciate. We hope to have improved the quality of our manuscript by revising it according to your comments. For detailed changes to the manuscript, please see your comments below:
The Introduction is excessively detailed for an original article. The focus of the study is missed and lacks a clear objective. Some references should be redistributed (in lines 51-52 the authors comment on variable sex and then again in line 61).
We agree that the introduction was formulated in too much detail. For this reason, we have revised the entire introduction and shortened it in various places.
“The screening phase was conducted according to the study protocol until a total of 20 athletes with elevated troponin I values above the upper reporting level (URL) and normal troponin T values could be identified. To do this, 219 athletes had to be included in the study”.
The authors should explain why they have selected this endpoint (20 athletes with cTnI > URL). Have you calculated the data size? Why not cTnT if in theory you did not know that that troponin was the one presenting elevation?
Many thanks for this important information. The authors had noticed an increased incidence of troponin I elevations caused by exercise stress from their own preliminary investigations. Therefore, the focus of this work was on the detection of troponin I in relation to troponin T. The initial assumption that in the case of 20 positive cases with troponin I elevation a statistically significant difference to troponin T was detectable was merely a working hypothesis and not part of the study protocol. In this respect, we have removed the relevant text passages from the manuscript and agree with the reviewer that this statement would not be statistically tenable.
If the design of the study consisted in including patients until you have 20 athletes with cTnI elevation, why in results you only have 18? Please comment on this.
As two athletes had to be excluded from the study due to incomplete data sets, the target number of troponin I positive athletes of 20 was reduced accordingly.
-“While, in contrast to patients with cardiovascular diseases, the mean troponin elevations in athletes after exercise are comparatively low and in the lower pathological range, the underlying causes of troponin elevation after exercise are still not clear.”
Results on Table 6 cannot be considered as low. These findings have to be further discussed.
A corresponding passage was added in the discussion, which deals with the extent of the troponin elevation found.
“The data presented in this paper lead to the conclusion that an exercise-induced increase in troponin I above the normal range is much more common in high-performance athletes during the training process than an increase in both troponins (I and T) or an increase in troponin T alone.”
The study design does not allow this conclusion to be reached.
A clear discussion of the authors’ results is missed.
We agree that a statement on the statistical significance of a more frequent troponin I than troponin T increase is not possible based on the data presented. The relevant text passage has therefore been revised and commented on accordingly.
-Line 58-59: Only Scherr et al [10] found higher exercise-induced cTnT values in the age group 29-51 years. Eijsvogels et al. suggested a relationship between …
The dot between years and Eijsvogels is missing.
Dot was filled in.
Line 62: difference for post exerciese cTnT,
Typo: I suggest not to use the abbreviation “postex.”
The corresponding passage in the introduction has been shortened. The abbreviation postex. has been changed throughout the text.
“Venous blood samples for plasma analysis were collected into 3,5ml Serum Plasma tubes,”
Please clarify if the tubes were serum or plasma. The manufacturer of the tube should be also declared.
The relevant text passage has been revised and the manufacturer of the tubes has been named.
Table 2 could be presented as Supplemental Material.
Table two was removed from the text and is now presented in the supplemental material section.
I recommend to use the abbreviations presented in line 50 along all the manuscript. In line 152 and Table 7, the authors use a new one (Trop T; Trop I)
All abbreviations for cTnI and cTnT have been standardised throughout the text.
Reviewer 2 Report
Comments and Suggestions for Authors
Thank you for giving me the chance to review this exciting paper. As far as I can see, this is the first paper that examines this problem, and I highly appreciate it. However, I do have a few concerns about it.
Firstly, the introduction is quite long and a little bit difficult to read. There are some punctuation and words missing, so it would be great to have a more specific introduction that clearly highlights why this paper is important.
Secondly, on page 4, line 125, there appears to be a mixing of fonts, which can be distracting to readers.
I find it interesting that almost all participants with elevated cTn were doing endurance exercise. In the follow-up of these participants, it would be useful to know whether they had any specific training prior to the blood sample. Although I understand that there is a lack of standardization, it would be helpful to have some idea of what kind of training was done prior to the test.
Additionally, some participants had really high values. Were they retested, and did they return to normal? It is necessary to provide some kind of follow-up information. Moreover, were any participants found to have cardiac disease?
Finally, it would be interesting to know whether any correlations were found between cTn and physical findings, such as heart rate, LV mass, etc.o normal?
Comments on the Quality of English Language
There are punctuation errors, missing words, and long, hard-to-understand sentences that need to be rewritten.
Author Response
Dear reviewer,
Thank you very much for your detailed and extremely helpful comments on our manuscript, which we greatly appreciate. We hope to have improved the quality of our manuscript by revising it according to your comments. For detailed changes to the manuscript, please see your comments below:
Firstly, the introduction is quite long and a little bit difficult to read. There are some punctuation and words missing, so it would be great to have a more specific introduction that clearly highlights why this paper is important.
We agree that the introduction was formulated in too much detail. For this reason, we have revised the entire introduction and shortened it in various places.
Secondly, on page 4, line 125, there appears to be a mixing of fonts, which can be distracting to readers.
The entire text has been revised and standardised concerning the fonts.
I find it interesting that almost all participants with elevated cTn were doing endurance exercise. In the follow-up of these participants, it would be useful to know whether they had any specific training prior to the blood sample. Although I understand that there is a lack of standardization, it would be helpful to have some idea of what kind of training was done prior to the test.
As already described in the manuscript and noted by you, it was not possible to standardise the pre-test conditions with regard to the training load.
A corresponding comment was added in the discussion section.
However, almost all of the endurance athletes were in the first training period of the current training year at the time of blood sampling, as the annual basic sports medicine examinations typically take place during this phase. This training period is largely characterised by a high proportion of low-intensity training with long training durations in all endurance sports. However, high-intensity training impulses are also occasionally used in this phase, so it is not possible to differentiate with certainty whether the proven troponin increases are caused by the low-intensity training with sometimes very long training periods or rather the high-intensity interval training.
Additionally, some participants had really high values. Were they retested, and did they return to normal? It is necessary to provide some kind of follow-up information. Moreover, were any participants found to have cardiac disease?
As already described in the text, no abnormalities were found in any of the athletes examined in the tests performed, nor did the athletes complain of any symptoms. In our preliminary work and subsequent observations, we followed selected athletes with significantly elevated troponin I levels over a longer period and were able to show a constant increase in troponin levels in all athletes. These investigations are part of a planned further publication and are therefore presented later..
Finally, it would be interesting to know whether any correlations were found between cTn and physical findings, such as heart rate, LV mass, etc.o normal?
The Spearman correlation analysis between the values of Troponin I and the other variables shows a slight to moderate negative correlation with Troponin I, but none of the p-values are low enough to indicate a statistically significant correlation, except for IVSd. IVSd shows a moderate negative correlation (-0.22) with Troponin I, and the p-value (0.0042) suggests a statistically significant correlation.
The Spearman correlation analysis between Troponin T and various variables predominantly showed non-significant correlations with slight to moderate negative or positive tendencies. Notably, body weight, LBM, and several other measurements indicated no correlation. A few variables, like body fat mass, exhibited a stronger positive correlation, yet the results were just above the threshold for statistical significance.
Overall, no clear or statistically significant relationships were established, indicating a generally weak association between Troponin T and the measured variables in this dataset. Due to the lack of statistically significant correlations, further elaboration on this matter in the paper is not pursued.
The above written text has been added to the manuscript.
There are punctuation errors, missing words, and long, hard-to-understand sentences that need to be rewritten
The entire text has been reviewed again for formal errors, and certain sentence structures have been simplified.
Round 2
Reviewer 1 Report
Comments and Suggestions for Authors
Dear authors,
thanks for considering my comments. The final version of the article is quite different than the first one you submitted.
I suggest you revise more carefully any other manuscript you write before submitting.
Yours sincerely
Reviewer 2 Report
Comments and Suggestions for Authors
No further comments